# Anhedonia in Relation to Reward and Effort Learning in Young People with Depression Symptoms

**DOI:** 10.3390/brainsci13020341

**Published:** 2023-02-17

**Authors:** Anna-Lena Frey, M. Siyabend Kaya, Irina Adeniyi, Ciara McCabe

**Affiliations:** 1School of Psychology and Clinical Language Sciences, University of Reading, Whiteknights Campus, Reading, RG6 6AL, UK; 2Department of Psychology, Abdullah Gül University, Kayseri 38080, Turkey

**Keywords:** anhedonia, depression, youth, learning, reward, effort

## Abstract

Anhedonia, a central depression symptom, is associated with impairments in reward processing. However, it is not well understood which sub-components of reward processing (anticipation, motivation, consummation, and learning) are impaired in association with anhedonia in depression. In particular, it is unclear how learning about different rewards and the effort needed to obtain them might be associated with anhedonia and depression symptoms. Therefore, we examined learning in young people (N = 132, mean age 20, range 17–25 yrs.) with a range of depression and anhedonia symptoms using a probabilistic instrumental learning task. The task required participants to learn which options to choose to maximize their reward outcomes across three conditions (chocolate taste, puppy images, or money) and to minimize the physical effort required to obtain the rewards. Additionally, we collected questionnaire measures of anticipatory and consummatory anhedonia, as well as subjective reports of “liking”, “wanting” and “willingness to exert effort” for the rewards used in the task. We found that as anticipatory anhedonia increased, subjective liking and wanting of rewards decreased. Moreover, higher anticipatory anhedonia was significantly associated with lower reward learning accuracy, and participants demonstrated significantly higher reward learning than effort learning accuracy. To our knowledge, this is the first study observing an association of anhedonia with reward liking, wanting, and learning when reward and effort learning are measured simultaneously. Our findings suggest an impaired ability to learn from rewarding outcomes could contribute to anhedonia in young people. Future longitudinal research is needed to confirm this and reveal the specific aspects of reward learning that predict anhedonia. These aspects could then be targeted by novel anhedonia interventions.

## 1. Introduction

Depression is a leading cause of illness and disability worldwide [1]. In order to develop new targets for treatment, a fundamental understanding of the mechanisms that underlie depression symptoms is needed. Anhedonia, the lack of interest and pleasure in normally rewarding experiences, is one of the main symptoms of depression and is thought to be underpinned by blunted reward responsiveness in adults [2,3,4,5] and adolescents [6,7,8]. Deficits in reward learning have also been observed in depression [9,10,11,12,13,14,15] and linked to anhedonia [16]. However, how anhedonia is related to effort learning and expenditure to obtain rewards is less well understood.

Studies examining effort-related processes in depression have found a lack of perseverance in depression, including a loss of motivation in the face of setbacks and failures [17], a reduced belief that one will conquer goals [18,19], negative perceptions of goal progress [20], and shifting beliefs about the value of rewards and the difficulty of achieving them [21]. Further, depressed patients exert less physical effort for monetary rewards compared to remitted individuals and controls [22,23,24], and show less willingness to engage in high effort to obtain high rewards compared to healthy controls [25].

Given that depression is one of the most common mental health problems in adolescence [26] which is associated with increased risk of suicide [27], long-term adverse health, economic, and social impacts [28], and high rates of recurrence during adulthood [29], it is imperative that we develop treatments that can intervene early in people’s lives to change depression’s negative trajectory. Further, as more than half of young people with depression demonstrate anhedonia symptoms at clinically significant levels [30], and anhedonia predicts poor treatment outcomes in adolescents [31], identifying novel targets for anhedonia treatment in young people is particularly important. Examining young people’s responses to reward and effort learning in relation to anhedonia may reveal such novel targets.

Our previous research on reward processing in adolescents has shown an association between higher anhedonia levels and less physical effort exertion for a primary reward (chocolate taste) in young people with depression symptoms [32]. However, as this study did not examine learning, it is as yet unclear how learning about rewards and the effort required to obtain them might be related to anhedonia in young people.

A recent study in adults examining reward preferences, physical and mental effort, and instrumental learning found that depressed adults provided higher aversive ratings for hypothetical effort (e.g., walking up 5 flights of stairs), had a higher tendency to avoid effort in a hypothetical yes/no choice task, and exerted less motor and cognitive effort compared to controls [33]. However, the study did not find any significant differences in reward and effort learning between depressed adults and controls, nor a significant relationship between depression or anhedonia symptoms and any of the task variables [33]. A possible explanation for this lack of a significant relationship observed in the study may be the small sample size (N = 20 for major depressive disorder). Consequently, it is yet to be established in a larger sample whether anhedonia is related to reward and effort learning in depression and how this might be represented in young people.

Therefore, the current study aimed to examine the relationship between anhedonia and learning about rewards and the effort required to obtain them in young people. In addition, we assessed the link between anhedonia and actual effort exertion, as well as subjective reports of willingness to exert effort and the “liking” and “wanting” of rewards. We recruited young people with a range of depression and anhedonia symptoms, as examining participants across a continuum of symptoms is regarded as more useful than group comparisons for identifying neurocognitive markers of depression [34]. We examined anhedonia using the Temporal Experience of Pleasure Scale, which has two subscales measuring anticipatory and consummatory anhedonia, respectively [35]. The participants completed a probabilistic reward and effort learning task adapted from Skvortsova et al. [36,37]. During the task, participants were asked to choose one of two shapes, each associated with a physical effort (high or low) leading to a reward (high or low). We adapted the task to include both primary (chocolate taste, images of puppies) and secondary (money) outcomes, to examine whether task responses generalize across different contexts, and because primary rewards may be more suitable for future investigations of reward and effort learning in younger adolescent populations. Before and after the task, participants were asked to provide subjective ratings of “liking”, “wanting” and “willingness to exert effort” for the rewards.

We hypothesized that anhedonia would be negatively correlated with reward liking, wanting and willingness to exert effort (as measured before the task) and actual effort exerted during the task. Further, we predicted a negative correlation between anhedonia and reward learning, but a positive correlation between anhedonia and effort learning, as Vinckier and colleagues (2022) [33] reported an elevated sensitivity to effort cost in depressed patients. Knowing if reward and effort learning are related to anhedonia symptoms could provide novel targets for treatment strategies for anhedonia. Moreover, identifying learning deficits in young people could help us to develop novel interventions for anhedonia, which are sorely needed.

## 2. Methods

### 2.1. Participants

As our main research question was focused on the relationship between learning and anhedonia symptoms, we based our sample size calculation on a (two-tailed) bivariate correlation test in G*Power. For this analysis with a medium effect size of 0.3, 80% power, and α = 0.05, a sample size of at least 84 participants was required. 

Young people (N = 132) between the ages of 17 and 25 years (mean age 20 yrs.) with a range of depression symptoms were recruited from the student population via the School of Psychology research panel, online advertisements, and posters throughout the university. Participants were assigned to one of three reward conditions: chocolate (N = 53), puppy pictures (N = 43), or money (N = 36). All participants were screened using an online version of the structured clinical interview for DSM-IV (SCID; adapted from [38]. The SCID was not used for diagnostic purposes, but to exclude those who ever experienced potentially clinical levels of symptoms of any Axis I disorder (besides depression or low levels of anxiety).

*All procedures contributing to this work comply with the ethical standards of the relevant national and institutional committees on human experimentation and with the Helsinki Declaration of 1975, as revised in 2008.* This study was reviewed and given a favorable ethical opinion for its conduct by the University of Reading Research Ethics Committee. After reading the information sheets, all participants provided informed consent. 

Participants were reimbursed for their time with course credits in the non-monetary reward conditions and with £10 in the monetary condition. All participants received a debriefing form, which advised anyone concerned about their mood to contact their GP and provided contact details for the Samaritans.

### 2.2. Procedure

#### 2.2.1. Questionnaires

After the screening, eligible participants filled out online versions of a demographics form, the Beck Depression Inventory—II (BDI—II; [39]), and the Temporal Experience of Pleasure Scale (TEPS, including anticipation and consummation subscales; [35]). 

High scores on the BDI indicate more severe depression symptoms. The questionnaire is widely used to assess depression symptoms, and its psychometric properties are well established and validated. 

Higher scores on the TEPS indicate fewer anhedonia symptoms, as this scale measures the experience of pleasure. The reliability of the TEPS subscales has been reported before [35] but is summarized again here. The TEPS items all positively intercorrelate, with a mean inter-item correlation for anticipatory and consummatory scales of *r* = 0.23 and 0.24, which falls within the recommended range [40]. The anticipatory and consummatory subscales showed good internal consistency (Cronbach’s *α* = 0.74 and 0.71, respectively), and the inter-correlation of the subscales was moderate at 0.41. The test–retest reliabilities for the subscales were all high, at r = 0.80 (*p* < 0.001), and 0.75 (*p* < 0.001), respectively. 

Subjects in the chocolate condition additionally completed the Eating Attitudes Test (EAT); [41]. An EAT score above 20 indicates a high level of concern about dieting, body weight, or problematic eating behaviors. 

After filling out the online questionnaires, participants attended a testing session during which they performed the learning task described below on a computer in the psychology department’s testing cubicles using the software MATLAB and PsychToolbox.

#### 2.2.2. Learning Task

We adapted a probabilistic instrumental learning task from the previous literature [36,37]. In addition to the monetary reward provided in the original task, we added a chocolate taste and photographs of puppies as high rewards and water and pictures of dogs as low rewards (see Figure 1A). During a pilot test and practice trials on the screening day, participants consistently rated ’liking’ higher for chocolate and puppies than for water and dogs, respectively. This is in line with observations in our previous studies using chocolate and with past findings that baby animals (e.g., puppies) are rated higher on several measures, including pleasantness, than adult animals (e.g., dogs) [42].

In order to complete the task, participants were seated in front of a desktop computer running Psychtoolbox (psychtoolbox.org, accessed on 1 June 2019), which is implemented within MATLAB (MathWorks, Natick, MA, USA). They registered their responses using a handheld dynamometer (Current Designs Inc., Philadelphia, PA, USA), which was calibrated to each individual. The force participants exerted was represented, in real time, by a green bar moving up and down within a rectangle outline drawn on the computer screen (see Figure 1B,C). The same calibration procedure as described by Skvortsova and colleagues [36] was used. In order to determine the maximal force a given participant was able to exert, subjects were asked to squeeze the handgrip as hard as they could for 15 s. Each participant’s maximal force (fmax) was calculated by taking the average of the data points that lay above the median force of that person. The force required during the task was individually adjusted to each participant’s maximal force by scaling the top of the rectangle outline to fmax. After the calibration trials and instructions, participants completed four practice trials. Subsequently, subjects were asked to rate the reward stimuli on a visual analogue scale ranging from 0 to 100. Specifically, they were asked to indicate how much they *liked* receiving the chocolate taste, looking at the puppy pictures, or receiving the money; how much they *wanted* to receive the chocolate taste, see the puppy pictures, or win the money; and finally, how much *effort they were willing to invest* to receive the chocolate taste, look at the puppies pictures or receive the money. These ratings were collected again at the end of the experiment.

During the tasks, every trial started with the option to choose between two shapes, shown on the left and right sides of the screen (see Figure 1C). Choices were made by pressing a key on the keyboard (z for the left option and m for the right option). Each option was associated with both a reward and a physical effort, but the left and right options differed in the level of the associated reward or effort. Effort and reward learning trials were interleaved, and each trial type was associated with one shape pair (i.e., the task included a total of two shape pairs/four shapes). On effort learning trials (shape pair 1), the effort level was probabilistic (80% vs. 20% high effort for the left vs. right option, sides counterbalanced), while the reward level was high for both options. On the reward learning trial (shape pair 2), the reward level was probabilistic (80% vs. 20% high reward for the left vs. right option, sides counterbalanced), while the effort level was high for both options. Participants were instructed to choose those options that resulted in receiving high rewards and/or avoiding high effort levels. The high effort level was set to 80% of fmax and the low effort level to 20% of fmax. Depending on the reward condition, high rewards were either a taste of chocolate, a picture of a puppy, or 50 p and low rewards were either a taste of water, a picture of a dog, or 10 p.

The tastes were delivered in 0.5 mL aliquots via long tubes to the participant, who held the tubes in their mouth. This amount is perceived as rewarding and prevents habituation, similar to our previous studies [32].

Once a choice was made, patients were informed about the outcome, i.e., the reward and effort levels were shown on the screen (reward: chocolate image, puppy line drawing, or money coin; effort: a visual target to reach within a black rectangle outline; see Figure 1). Next, the rectangle outline turned white, signaling to participants to start exerting effort. Participants were required to squeeze the handgrip until the green bar reached the target (horizontal line) and to hold the target force for 1 s. Any force participants exerted beyond the target was not shown on the screen but was recorded to determine the maximum force invested on that trial. The two behavioral responses (choice and force exertion) were self-paced. Therefore, participants needed to produce the required force to proceed further, which they managed to achieve on every trial. After participants reached the effort target, they received the actual reward (either a taste of chocolate or water, seeing a photograph of a puppy or dog, or a 50 p or 10 p coin).

Overall, the task consisted of 50 trials (17 trials, short break, 17 trials, short break, 16 trials) and took about 40 min to complete. The trial number was chosen to provide participants with enough repetitions to learn the contingencies, while limiting the potentially detrimental effects of fatigue if the task was too long.

The data generated consisted of ratings of “liking,” “wanting,” and “willingness to exert effort” for rewards (collected at the start and the end of the task), task reward and effort learning accuracy, effort completion times (started from the first initiation of the squeezing); and effort force (the maximum amount of force the participants exerted on each trial, expressed as a percentage of their maximum effort).

### 2.3. Analysis

All data were examined using Excel and SPSS. In line with a dimensional approach, Pearson’s correlations were performed to examine the relationship between anhedonia scores (anticipatory and consummatory TEPS subscales) and the subjective ratings of wanting, liking, and willingness to exert effort before the task and task performance. 

To examine whether responses differed between reward conditions, which would suggest a lack of generalization across contexts, we conducted mixed measures ANOVAs with reward condition (money, puppies, chocolate) as the between subject factor, and subjective ratings and task measures as within-subject factors. 

Visual inspection of the task data, using box-and-whisker plots, revealed several clear outliers in the effort completion times and the effort force data. Therefore, values outside ±2 standard deviations of the mean were removed from these datasets.

## 3. Results

### 3.1. Demographics and Questionnaire Measures

Table 1 describes the study’s demographics. Participants had a mean age of 20 years and a broad range of depression and anhedonia symptoms.

### 3.2. Trait Anhedonia and Subjective Liking, Wanting and Willingness to Exert Effort

Firstly, to check for habituation effects on the stimuli, we examined changes in subjective ratings from the beginning to the end of the task. We used mixed-measure ANOVAs with condition (money, puppies, chocolate) as the between subject factor, and rating type (wanting, liking, willingness to exert effort) and time (before and after task completion) as the within-subject factors. We found no main effect of time, nor any interactions with time (all *p* > 0.05).

Secondly, we examined the relationship of anticipatory and consummatory anhedonia (TEPS-A and TEPS-C) to the subjective ratings of “liking”, “wanting” and “willingness to exert effort” collected at the beginning of the task, after the practice trials. Using partial Pearson’s correlation and controlling for depression (BDI), we found that those with higher anticipatory anhedonia (indicated by lower scores on TEPS A) had lower liking (*r* = 0.22, *p* = 0.008) and wanting ratings (*r* = 0.20, *p* = 0.012; see Figure 2). When participants with an EAT score above 20 (N = 4) were removed from the analysis, the correlations remained significant for liking (*r* = 0.25, *p* = 0.002) and wanting ratings (*r* = 0.23, *p* = 0.006), and a correlation between TEPS A and willingness to exert effort emerged (*r* = 0.14, *p* = 0.056). No other significant relationships were found.

### 3.3. Trait Anhedonia and Task Measures

Next, we examined the relationships between trait anhedonia and reward and effort learning, as well as between anhedonia and effort exerted during the task. Using partial Pearson’s correlations controlling for BDI, we found that those with higher anticipatory anhedonia (lower scores on TEPS A) demonstrated lower reward learning accuracy (*r* = 0.21, *p* = 0.009; see Figure 3), while no significant relation between anhedonia and effort learning accuracy was observed (*r* = 0.04, *p* = 0.339). There were no significant relationships between effort or reward learning accuracy and consummatory anhedonia (TEPS C) or between anhedonia and effort exerted during the task. When those with an EAT score above 20 (N = 4) were removed from the analysis, the results remained the same for reward (*r* = 0.25, *p* = 0.003) and effort learning (*r* =0.05, *p* =0.287) accuracy, but a correlation between low effort completion time and TEPS C emerged (*r* = −0.16, *p* = 0.04). Note that after application of the Benjamini-Hochberg correction for multiple comparisons (with a false discovery rate of 10% [43]), all the correlations remained significant.

### 3.4. Differences between Trial Types and Reward Conditions

In order to examine whether responses differed between reward conditions, which would suggest a lack of generalization across contexts, we conducted mixed measures ANOVAs with reward condition (money, puppies, chocolate) as the between subject factor, and the following variables as within-subject factors (all examined in separate ANOVAs): subjective ratings (wanting, liking, willingness to exert effort before the task), maximum effort exertion (in high and low effort trials), effort completion times (in high and low effort trials), and learning accuracy (in reward and effort learning trials).

For the subjective ratings, we found a main effect of rating type (*F*(1.69, 213) = 12.85, *p =* 0.001) and a rating type by reward condition interaction (*F*(3.38, 213) = 2.79, *p =* 0.035), with higher willingness to exert effort for money than chocolate (*t*(87) = −2.79; *p* = 0.006) or puppies (*t*(75) = 2.15; *p* = 0.034), and higher liking (*t*(93) = −1.98; *p* = 0.050) and wanting (*t*(92) = −2.44; *p* = 0.017) for puppies than chocolate (see Figure 4). 

For learning, we found a main effect of learning trial type (*F*(1,129) = 12.31, *p =* 0.001), with better reward than effort learning (see Figure 5), but no significant main effect of condition or learning by condition interaction.

Moreover, for maximum effort exertion (/force), we found the expected main effect of effort level (*F*(1,125) = 900, *p* < 0.001), with more effort exerted on the high vs. low effort trials. We also found a main effect of reward condition (*F*(2,125) = 5.98, *p =* 0.003). A follow-up ANOVA with Bonferroni-corrected post-hoc tests revealed that this effect was driven by more effort exertion in the money condition than in the chocolate condition (*p* = 0.007).

Further, when examining effort completion times, we found the expected main effect of effort level (*F*(1,122) = 279.35, *p* < 0.001), with high effort trials taking longer to complete than low effort trials. No significant main effect of condition or effort by condition interaction was observed. The results remained the same when those with a high EAT score (N = 4) were removed.

## 4. Discussion

The main aim of this study was to examine the relationship of anhedonia to subjective and objective measures of reward and effort processing sub-components in young people with a range of depression symptoms. The secondary aim was to assess, using novel primary (chocolate and puppies) and secondary (money) rewards, whether observed responses would generalize across different contexts.

Firstly, we observed no significant effect of time on subjective ratings collected before and after the task. This provides a tentative indication that there was likely no habituation to the rewards during the task (although no definite conclusion can be drawn from non-significant results). This supports novel primary rewards (chocolate and puppy images) as suitable for assessing reward- and effort-related behavior over time.

Further, as hypothesized, subjective liking and wanting of rewards decreased as anticipatory anhedonia increased. This finding is consistent with our recent study in which young people with depression symptoms described feelings of reduced pleasure and motivation when interviewed about their experiences of anhedonia [44].

When examining learning, we found that participants made significantly more correct choices (i.e., demonstrated higher accuracy) on reward learning trails than on effort learning trials. This finding is similar to the observations of Skvortsova and colleagues [37,38], who also reported higher accuracy for reward vs. effort learning, although their finding did not reach significance. A possible reason why we observed a significant effect of trial type on learning accuracy may be the inclusion of primary rewards, which may have made the reward outcomes particularly salient [45]. This may have shifted participants’ attention/cognitive resources towards reward-based learning rather than effort-based learning. In addition, the fact that young people are better able to learn from rewarding outcomes than from negative ones, such as effort [46], may also have contributed to our findings. Larger studies with adults are needed to establish robust effort learning behaviors which can then be related to anhedonia measures.

We further observed, as hypothesized, that higher anticipatory anhedonia was associated with lower reward learning accuracy. This finding is consistent with the literature on dysfunctional learning processes in patients with major depressive disorder [47].

Our finding, however, extends previous work by explicitly linking lower reward learning accuracy with increased anhedonia symptoms in young people. As reward learning is known to predict depression outcomes in adults and young people [7,48], the implication of this work is that anhedonia, too, may be predicted by deficits in reward learning. Future studies could test this by examining, with longitudinal designs, whether reward learning deficits predict subsequent emergence of anhedonia symptoms. If this is found to be the case, this would have clinical implications, as reward learning could serve as a novel target for anhedonia treatment and prevention.

In conclusion, our study provides new insights into the relation between anhedonia and reward learning in young people and underscores the importance of examining factors that relate to anhedonia, which may offer novel treatment targets. In addition, we demonstrate the feasibility of examining learning with the use of primary rewards, which may be particularly salient and suitable for younger adolescents.

## Figures and Tables

**Figure 1 brainsci-13-00341-f001:**
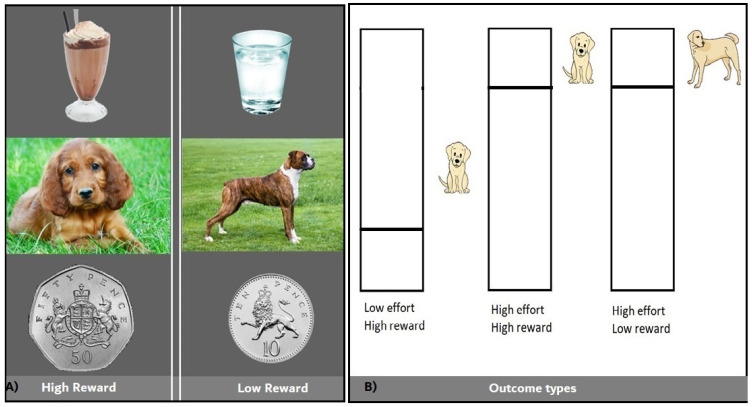
(**A**) Reward conditions: High Reward = Chocolate, Low Reward = Water; High Reward = Puppies, Low Reward = dogs; High Reward = 50 p, Low Reward = 10 p. (**B**) Outcome types: Low Effort for High Reward; High Effort for High Reward; High Effort for Low Reward. (**C**) Example High Reward (chocolate) for High Effort trial.

**Figure 2 brainsci-13-00341-f002:**
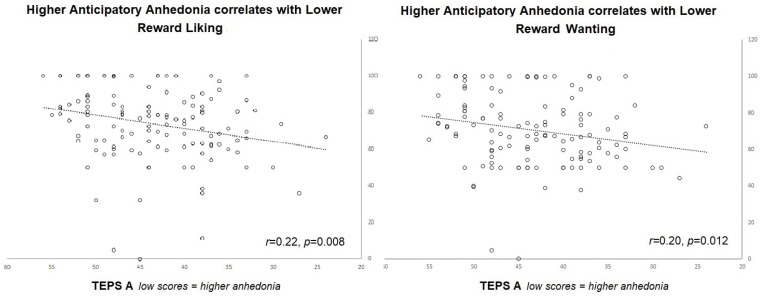
Shows subjective ratings of liking and wanting plotted against anhedonia scores (TEPS-A).

**Figure 3 brainsci-13-00341-f003:**
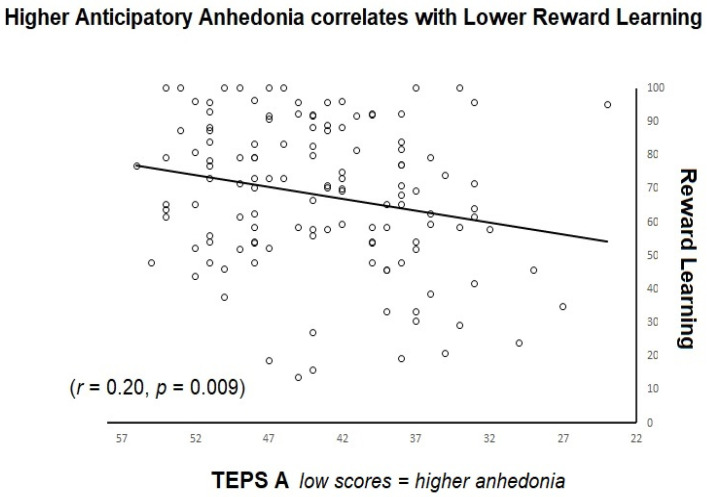
Shows reward learning accuracy plotted against anticipatory anhedonia scores (TEPS-A).

**Figure 4 brainsci-13-00341-f004:**
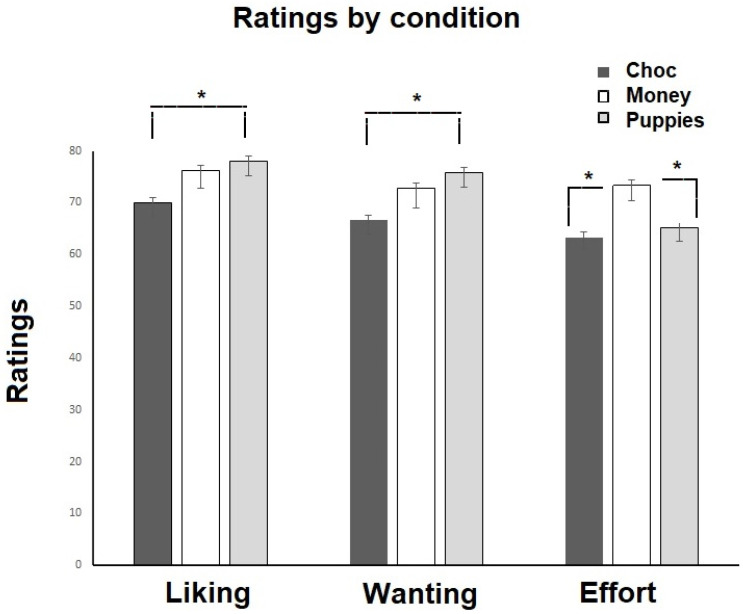
Shows the subjective ratings for liking, wanting and willingness to exert effort for each condition (choc: chocolate, money and puppies). * indicates significant differences.

**Figure 5 brainsci-13-00341-f005:**
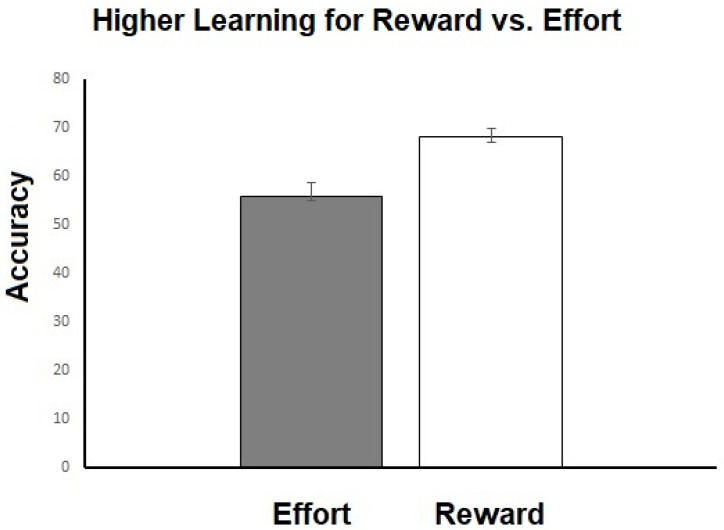
Shows mean accuracy for reward and effort learning across all participants.

**Table 1 brainsci-13-00341-t001:** Demographic and questionnaire measures.

	Mean	SD
Age	20.64	2.74
BDI	11.95	10.164
EAT	8.41	8.037
TEPS_A	43.50	6.759
TEPS_C	35.53	6.140

BDI, Beck Depression Inventory; EAT, Eating Attitudes questionnaire; TEPS_A, Temporal anticipatory experience of pleasure scale; TEPS_C, Temporal consummatory experience of pleasure scale; SD, Standard Deviation.

## Data Availability

The data are not publicly available due to lack of consent to share the data publicly. The data presented in this study are available on request from the corresponding author for researchers who meet the criteria for access to confidential, de-identified data.

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
