# Peer review of "Anhedonia in Relation to Reward and Effort Learning in Young People with Depression Symptoms"

_brainsci, 2023, doi:10.3390/brainsci13020341_

Round 1

Reviewer 1 Report

The authors examined the relationship between anhedonia and reward and effort learning among young people with depression symptoms. Overall, the manuscript is well-written and I only have several minor comments:

1) Abstract: The objective is not clear.

2) Introduction: Please describe the two types of anhedonia measured in this study.

3) Participants: Since the target participants are young people with depression symptoms, what are the inclusion criteria?

4) 2.2.1. Questionnaires: Kindly report the reliability of all the (sub)scales.

5) Discussion: Please discuss the implications more (especially in terms of clinical settings) as well as limitations and suggestions for future research.

Author Response

Reviewer 1:

1) Abstract: The objective is not clear.

-- We have now made this clear in the abstract.

2) Introduction: Please describe the two types of anhedonia measured in this study.

--This has now been added to the intro

3) Participants: Since the target participants are young people with depression symptoms, what are the inclusion criteria?

--We have clarified this in the methods section “Young people (N=132) between the age of 17 and 25 years (mean age 20 yrs.) with a range of depression symptoms were recruited from the student population via the School of Psychology research panel, online advertisements and posters throughout the university.”

4) 2.2.1. Questionnaires: Kindly report the reliability of all the (sub)scales.

--We have added reliability from the original paper on the TEPS development Gard et a., 2007.

5) Discussion: Please discuss the implications more (especially in terms of clinical settings) as well as

limitations and suggestions for future research.

  • We have added some more on the implications and future directions.

Reviewer 2 Report

Thank you for offering me the opportunity to review this manuscript. The paper focused on clarifying the associations between anhedonia and effort and reward learning processes in young adults. Although the general idea behind the analysis is good, the exhibition of it is poor. Further explanations are provided below to help revising the manuscript.

Abstract

·     1.   The abstract provides a nice and general summary of the study, but it is not completely consistent with the information provided in the main text. For example, the authors mentioned that “longitudinal research” is needed but this is not mentioned in the “Discussion” section. Further revisions are needed in the main text to achieve this.  

Introduction

·    2.   The introduction provides a nice summary of previous evidence but lacks explanation of the practical implications of this study and why this study provides important evidence that it is indeed missing from our understanding of depression.

·     3. The Introduction will benefit from a more thorough explanation why the authors focused on young adults for this study.

·     4.   P.2, line 50 – it should be “healthy” controls and not “health” controls.

·     5.   The reference to “young adolescents” in line 81 is a bit confusing in regards to the sample recruited, which were more young adults.

Methods

·     6. The authors mentioned that previous studies had small sample sizes, which inhibited them from finding significant findings. Did the authors calculate the required sample for this study a priori to ensure that they had the power needed to find significant findings? For example, was G*Power used?

·     7. "2.2.1 Questionnaires": I understand that the questionnaires used are quite famous, however, when writing a paper it is always nice to write it in a way that it is comprehensible to scientists working outside the topic of the paper (as mentioned in the author guidelines of the journal). Therefore, this section would definitely benefit from further clarifications on the questionnaires used, e.g. a brief description on the number of items, what they are investigating, what higher scores mean and if they are validated or not. It is difficult for someone who is not familiar with the questionnaires to understand the means provided in Table 1 and what they represent without this information in the “2.2.1 Questionnaires” section.  

·     8.   "2.3 Analysis": It is not clear what the authors mean by “in line with a dimensional approach, Pearson’s...” Could the authors further explain this or just remove this?

·      9. The sentence in line 193-194: “All data were examined using Excel and SPSS” should be moved to the “Analysis” section.

·     10.   The authors mention that they used Pearson’s correlations, but later on in the result section they report mixed-measure ANOVAs. The analysis section would benefit from a more elaborative/inclusive description of what was actually done for the analysis of the outcomes, including whether any normality tests were conducted.

Results

·     11.   Although the EAT questionnaire was administered to those in the chocolate condition – the outcomes of the EAT were not accounted for in any of the relevant analyses.

Discussion

·      12. The authors suggest that the outcomes of this study might be useful for novel treatment targets/developments. The discussion would benefit from more concrete examples of what the authors want to say.

·      13. The direction of the relationship between anhedonia and reward learning becomes a bit confusing in the “Discussion” section. Do the authors suggest that a biderectional relationship might be possible? Please clarify.

Author Response

Abstract

  •    1.   The abstract provides a nice and general summary of the study, but it is not completely consistent with the information provided in the main text. For example, the authors mentioned that “longitudinal research” is needed but this is not mentioned in the “Discussion” section. Further revisions are needed in the main text to achieve this.  

-- We did mention running studies to examine reward learning predicting anhedonia in the discussion but it wasn’t as clear as it could have been so we have corrected this.

Introduction

  •   2.   The introduction provides a nice summary of previous evidence but lacks explanation of the practical implications of this study and why this study provides important evidence that it is indeed missing from our understanding of depression.

--We agree and added a sentence at the end of the intro to make this clear.

  •    3. The Introduction will benefit from a more thorough explanation why the authors focused on young adults for this study.

--This is been amended in the intro

  •    4.   P.2, line 50 – it should be “healthy” controls and not “health” controls.

--corrected.

  •    5.   The reference to “young adolescents” in line 81 is a bit confusing in regards to the sample recruited, which were more young adults.
  • This has been changed to young people

Methods

  •    6. The authors mentioned that previous studies had small sample sizes, which inhibited them from finding significant findings. Did the authors calculate the required sample for this study a priori to ensure that they had the power needed to find significant findings? For example, was G*Power used?

--Yes we have done this and added a section on gpower now in the methods

  •    7. "2.2.1 Questionnaires": I understand that the questionnaires used are quite famous, however, when writing a paper it is always nice to write it in a way that it is comprehensible to scientists working outside the topic of the paper (as mentioned in the author guidelines of the journal). Therefore, this section would definitely benefit from further clarifications on the questionnaires used, e.g. a brief description on the number of items, what they are investigating, what higher scores mean and if they are validated or not. It is difficult for someone who is not familiar with the questionnaires to understand the means provided in Table 1 and what they represent without this information in the “2.2.1 Questionnaires” section.  

--we had now added more on this to the paper

  •    8.   "2.3 Analysis": It is not clear what the authors mean by “in line with a dimensional approach, Pearson’s...” Could the authors further explain this or just remove this?

--This is in relation to examining anhedonia as a continuum not as high vs low

  •     9. The sentence in line 193-194: “All data were examined using Excel and SPSS” should be moved to the “Analysis” section.

--done

  •    10.   The authors mention that they used Pearson’s correlations, but later on in the result section they report mixed-measure ANOVAs. The analysis section would benefit from a more elaborative/inclusive description of what was actually done for the analysis of the outcomes, including whether any normality tests were conducted.

--this has been made more clear and mixed measures added to analysis section.

Results

  •    11.   Although the EAT questionnaire was administered to those in the chocolate condition – the outcomes of the EAT were not accounted for in any of the relevant analyses.

 --We have added this now to the paper, only 4 participants had an EAT score over 20 which indicates considerable worry about food, so we ran the analysis again without those participants and the data remained the same.

Discussion

  •     12. The authors suggest that the outcomes of this study might be useful for novel treatment targets/developments. The discussion would benefit from more concrete examples of what the authors want to say.

--we have added more on implications in discussion

  •     13. The direction of the relationship between anhedonia and reward learning becomes a bit confusing in the “Discussion” section. Do the authors suggest that a biderectional relationship might be possible? Please clarify.

--We hope the rewritten sections make it now clear, reward learning could predict anhedonia

Reviewer 3 Report

Intro, line 30 - is this true, or does the word "mental" need to be inserted?

Line 207, legend: "TEPS-C" is incorrectly named.

Paper could benefit from adding a conclusions section. I had to answer "not applicable" about whether the conclusions were supported by the results because, at least formally, no set of conclusions is set off from the discussion.

Expecting that such conclusions would include a statement that reward learning should be evaluated (further?) for efficacy in improving depression, a statement about what is known about this should be added to the discussion. 

Author Response

Intro, line 30 - is this true, or does the word "mental" need to be inserted?

--corrected to “a” leading cause stated by WHO

Line 207, legend: "TEPS-C" is incorrectly named.

--fixed

Paper could benefit from adding a conclusions section. I had to answer "not applicable" about whether the conclusions were supported by the results because, at least formally, no set of conclusions is set off from the discussion.

--we have added a conclusion with implications of this work

Expecting that such conclusions would include a statement that reward learning should be evaluated (further?) for efficacy in improving depression, a statement about what is known about this should be added to the discussion. 

--we have rewritten the end of the discussion

Round 2

Reviewer 2 Report

Thank you for revising the manuscript based on the comments provided. All the comments were addressed satisfactorily. 

I would only suggest one very minor typo revision in line 77 for the word "people".